# Two-Step Dopamine-to-Polydopamine Modification of Polyethersulfone Ultrafiltration Membrane for Enhancing Anti-Fouling and Ultraviolet Resistant Properties

**DOI:** 10.3390/polym12092051

**Published:** 2020-09-09

**Authors:** Sri Mulyati, Syawaliah Muchtar, Nasrul Arahman, Yanna Syamsuddin, Normi Izati Mat Nawi, Noorfidza Yub Harun, Muhammad Roil Bilad, Yuliar Firdaus, Ryosuke Takagi, Hideto Matsuyama

**Affiliations:** 1Department of Chemical Engineering, Universitas Syiah Kuala, Banda Aceh 23111, Indonesia; syawaliah2009@gmail.com (S.M.); nasrular@unsyiah.ac.id (N.A.); yanna_syamsuddin@unsyiah.ac.id (Y.S.); 2Department of Chemical Engineering, Universiti Teknologi PETRONAS, Bandar Seri Iskandar 32610, Perak, Malaysia; normi_16000457@utp.edu.my (N.I.M.N.); noorfidza.yub@utp.edu.my (N.Y.H.); 3Division of Physical Sciences and Engineering (PSE), KAUST Solar Center (KSC), King Abdullah University of Science and Technology (KAUST), Thuwal 23955-6900, Saudi Arabia; yuliar.firdaus@kaust.edu.sa; 4Research Center for Membrane and Film Technology, Department of Chemical Science and Engineering, Kobe University, Rokkodai-Cho 1-1, Nadaku, Kobe 657-8501, Japan; takagi@harbor.kobe-u.ac.jp (R.T.); matuyama@kobe-u.ac.jp (H.M.)

**Keywords:** polydopamine modified membrane, anti-ultraviolet, anti-fouling, polymerization, polyethersulfone

## Abstract

Polydopamine has been widely used as an additive to enhance membrane fouling resistance. This study reports the effects of two-step dopamine-to-polydopamine modification on the permeation, antifouling, and potential anti-UV properties of polyethersulfone (PES)-based ultrafiltration membranes. The modification was performed through a two-step mechanism: adding the dopamine additive followed by immersion into Tris-HCl solution to allow polymerization of dopamine into polydopamine (PDA). The results reveal that the step of treatment, the concentration of dopamine in the first step, and the duration of dipping in the Tris solution in the second step affect the properties of the resulting membranes. Higher dopamine loadings improve the pure water flux (PWF) by more than threefold (15 vs. 50 L/m^2^·h). The extended dipping period in the Tris alkaline buffer leads to an overgrowth of the PDA layer that partly covers the surface pores which lowers the PWF. The presence of dopamine or polydopamine enhances the hydrophilicity due to the enrichment of hydrophilic catechol moieties which leads to better anti-fouling. Moreover, the polydopamine film also improves the membrane resistance to UV irradiation by minimizing photodegradation’s occurrence.

## 1. Introduction

Polydopamine (PDA) has been known as an important biocomponent in the medical field, material coating, and an effective additive for membrane fabrication [1,2,3]. PDA is bio-inspired material that can be easily formed under mild conditions (i.e., room temperature) [4] on almost all inorganic and organic substrates [1,5]. The abundant functional groups from catechol on PDA allow it to be a versatile platform for further adhesion of other components with the desired function [6].

The introduction of the PDA as a material modifier was inspired by the mussel foot with a strong adhesive ability [5,7]. PDA is imposed onto a substrate via surface coating thanks to the rapid formation of the PDA layer through polymerization, triggered by an autooxidation reaction using a basic precursor [8]. Such a polymerization mechanism is favored as the other components that can be functionalized on the PDA-layer adhered strongly to the substrate [8,9,10,11].

By its attractive and beneficial characteristics, PDA has been progressively used in many fields [12,13]. PDA has been used as an additive in membrane fabrication [5,14,15,16], thanks to its very high degree of hydrophilicity attributed to the amine and hydroxyl-rich groups. Incorporation of PDA in a membrane matrix then increases the hydrophilic trait that is beneficial for inhibiting the membrane fouling [14,17,18].

In membrane fabrication, numerous surface modification techniques such as surface patterning [19,20,21], polymer blending [17,22,23,24], and surface coating [25] have been applied to enhance the membrane properties in reducing its fouling propensity. The surface coating method is popular to fabricate membranes used in a photocatalytic reactor, in which PDA can act as a medium for binding photocatalyst components that otherwise cannot be deposited directly on the membrane surface [26]. In addition, the presence of a PDA layer on the photocatalytic membrane protects the membrane against UV radical attacks and minimizes the photodegradation [26,27,28]. Nonetheless, PDA’s incorporation also decreases permeation due to blocking the pore surface by the PDA layer [28]. Meanwhile, if the PDA blended with the polymer, the photocatalytic efficacy diminishes because the photocatalysis occurs only on the surface [27]. Another employed a simple dip-coating in an aqueous solution of dopamine and utilized dopamine self-polymerization to form thin surface-adherent polydopamine films onto various materials in which secondary reactions can be used to create a variety of ad-layer [29]. Recent reviews on surface functionalization’s, including for membrane modifications, have been reported elsewhere [30,31].

In our previous work [32], we adopted a new technique for modifying PDA on PDVF through a combination of blending and polymerization that resulted in high permeation, humic acid (HA) rejection, and antifouling performance. As an additive, dopamine acts as a pore former, which synergistically promoted the water flux without rejection trade-off. However, direct PDA coating resulted in membranes with a rather poor permeability. The dopamine blending approach was explored to utilize its pore forming role that enhances permeability and hydrophilicity. Coating of the membrane directly with PDA would require additional pore former. Therefore, a combination of the both methods was expected to simultaneously improve membrane filterability, antifouling, and anti UV performance. With the intention of further investigating dopamine modification techniques, herein, the same modification method is attempted on the PES membrane but with more variations in the concentration of dopamine, and it is aimed at the resistance of the membrane to deeper UV exposure. Anti-UV studies of PDA-modified membranes through this technique are reported for the first time in this work. It is expected that the proposed fabrication technique not only resulted in membranes with outstanding permeability, selectivity, and antifouling performances, but also excellent UV resistance; as such, it can be further applied as a photocatalytic membrane. The presence of the PDA layer on the membrane surface can be used to functionalize semiconductor photocatalyst particles such as TiO_2_ and so on. This study focuses on the role of PDA additive as an anti-fouling agent and anti UV degradation agent and high filtration performance. Such properties are important for membrane materials used in photocatalytic membrane reactors because the presence of the PDA layer can be functionalized with semiconductor photocatalytic particles.

## 2. Materials and Methods

### 2.1. Materials

Polyethersulfone (PES, BASF, Ludwigshafen, Germany) was used as a base polymer for membrane materials. N-methyl pyrrolidone (NMP, Merck, Hohenbrun, Germany) was used as a solvent. Dopamine hydrochloride (Sigma Aldrich, MO, USA) and Tris (2-Amino-2-hydroxymethyl-1,3-propanediol) (Sigma Aldrich, MO, USA) were employed as an additive and buffer solutions, respectively. HA (Sigma Aldrich, MO, USA) was used as a foulant model for the filtration experiments. Distillate water was used as a non-solvent for membranes, solvent for the Tris-buffer preparation, and as the feed for the pure water filtration experiments.

### 2.2. Preparation of Membrane

The membrane preparation utilized the phase inversion technique, illustrated in Figure 1. The dope solutions were prepared by dissolving PES with the dopamine of varied concentrations (0.5(a); 2(b); and 4(c)%) in the NMP. The mixtures were stirred until homogeneous at room temperature. Once homogeneous, the solution was cast on top of a glass plate at a wet thickness of 2 mm thick film by using the knifing stick. The formation of the membrane sheet occurred after the cast film was immersed in a water container. Some of the membranes that were ready were coded as P-D_0_ and kept in DI water until used further. The remaining membranes were dipped into a base Tris-HCl solution (pH: 8.5) with a varying dipping duration of 5, 24, and 36 h. Pristine PES membrane (P) without the addition of dopamine and dipping treatment was also prepared as the control. The summary of dope solution formulations for all membrane samples is given in Table 1.

### 2.3. Membrane Characterization

Several characterizations were conducted to study the effect of the modification on the resulting membranes. The FTIR Spectrophotometer was employed to distinguish the change in chemical groups of the membranes (Thermo Scientific iD5 ATR-Nicolet iS5, Tokyo, Japan). A field-emission scanning electron microscope (FE-SEM, JSF-7500F, Jeol Co. Ltd., Tokyo, Japan) was employed to observe the structural morphologies after modification and after UV irradiation. The membranes’ mechanical strength was measured by pulling the membrane sample in a dumbbell shape with dimensions of 4 mm width × 40 mm long using Autograph AGS-J (Shimadzu Co., Kyoto, Japan). The hydrophilicity degree of the membranes was investigated by measuring the surface water contact-angle measurement using Drop Master 300 (Kyowa Interface Science Co., Niiza-City, Japan). The thickness and porosity of the membrane samples were measured using a micrometer and dry/wet method, respectively. Due to the dense nature of the resulting membranes, the pore size was estimated based on the molecular size of the HA applied in this study as reported elsewhere [33].

### 2.4. UV Irradiation Test

A membrane piece was put on flat glass and irradiated by a UV lamp (22 W, 254 nm SUV-16, AS ONE, Nishi-ku, Japan) with a membrane-to-lamp gap of 5 cm. The irradiation experiment was conducted in a tightly closed box for a total of 7 days.

### 2.5. Filtration Test

The filtration tests were conducted using a UF cell at a pressure of 1.5 bar. This experiment was done using two feeds (DI water and 50 ppm HA solution) to evaluate the performance of membranes in terms of pure water flux (PWF), HA solution flux, and rejection of HA solution. This experiment’s procedure was carried out in one set by changing feed from DI water to HA solution followed by membrane cleaning. It continued with replacing the feed with DI water again to obtain the data for antifouling assessment. The detailed information on this experimental procedure has been described in our earlier works [28].

## 3. Results and Discussion

### 3.1. Residual Dopamine or Polydopamine

The FTIR results in Figure 2 confirm that the residual dopamine presented in the membrane matrix. The analysis shown here is only for pristine PES (P), the membrane with low dopamine concentration and short polymerization time (P-D5-a), high dopamine concentration and short polymerization time (P-D5-c), and high concentration and long polymerization time (P-D36-c). It is distinguished by the catechol-amine moiety, specific to dopamine, appearing at the spectrum range of 3300–3600 cm^−1^. They arise due to the stretching effect of the NH bond as well as from the hydroxyl group of alcohol [34]. Though weaker in intensity, another peak transmission is also detected in the spectra of modified membranes, a wavenumber of 1640 cm^−1^ [28,32,34]. The emergence of this peak is the sign of NH vibration from dopamine/polydopamine.

### 3.2. Hydrophilicity

Water contact angle data in Figure 3 demonstrate that the addition of dopamine at higher concentrations and longer dipping times improves the hydrophilicity of the resulting membranes, as can be seen from the decrement of the water contact angle. The hydrophilic membrane has an excellent wetting characteristic indicated by the better spreading of the water droplet on the membrane surface and forming a low water contact angle [35]. The improved hydrophilicity of these membranes was greatly influenced by the presence of catecholamine groups (N-H and O-H) in the PDA layer, which helped weaken the interaction between the surface and the foulants to impose anti-fouling properties [8,36]. Moreover, the addition of hydrophilic dopamine particles could have improved the pore characteristics of the modified membranes, as demonstrated in our earlier report [29].

### 3.3. Hydraulic Performance

The PWF data of the prepared membranes are shown in Figure 4. It shows that the non-modified PES membrane (P) poses a low PWF of 15 L/m^2^.h because of the membrane’s non-porous and hydrophobic nature. For P-D5-a, P-D5-b, and P-D5-c membranes, the PWF improves by threefold, presumably attributable to the enhancement of membrane characteristics in terms of porosity. However, if the membrane polymerized for more than 24 h, the overgrowing of the PDA film via auto-oxidation triggered by the alkaline Tris-HCl solution leads to a decrease in PWF. Although beneficial in terms of hydrophilicity, the PDA film blocks the membrane pores, especially those on the surface, which adds hydraulic resistance that lowers water flux [37].

Figure 5 shows that an increase in the flux of HA solution is observed in almost all modified membranes. The influences of dopamine addition at various concentrations and dipping times to the water flux and rejection of HA solution were monitored through the ultrafiltration test. This increase is generally caused by an increase in membrane porosity, as detailed in our earlier report [29]. In contrast, a very slim decrease in water flux is detected in the HA water flux of the P-D36-a membrane as the dopamine concentration is relatively low (0.5%), resulting in almost no effect on the pore formation. Moreover, 36 h of dipping in the Tris solution allowed the thicker PDA layer growth on the membrane, which mostly covered the surface of the P-D36-a membrane.

The pore size of the membrane samples could not be detected by the capillary flow porometer because, for a pressure limit of maximum 200 bar, correspond ing to minimum pore size of about 0.05 µm. By considering the HA rejection, we concluded that all of the membarnes samples were tight UF. The conclusion was made by estimating the minimum size of the HA, which could be very well rejected by all membrane samples (with humic acid rejection of >60%). The applied HA had a molecular weight of 226 Da, corresponding to a minimum diameter of 0.8 nm. From a relatively small variation of HA rejection, all membranes seem to have a similar pore size, suggesting that dopamine concentration and polymerization time did not significantly affect the pore size.

For HA rejection, generally, it is seen that the dopamine-blended membranes generate a permeate with a slightly lower rejection than that of the pristine PES membrane. The pristine PES membrane poses the highest HA rejection of 87% due to its dense structure. The decrease in HA rejection value is attributed to the increase of surface porosity and pore size of the membrane, which is discussed later. Allegedly, some HA particles whose size is in the nanometer scale escape through the membrane pores. However, the polymerized P-D24 and P-D36 membranes at dopamine concentrations of 0.5%, 2%, and 4% experience a slight increase in rejection because the membrane surface is coated by PDA film due to polymerization that partly retained the HA particles.

### 3.4. Fouling Resistance

Figure 6 demonstrates that dopamine-modified membranes exhibit better antifouling properties than the pure PES membrane in terms of total flux loss due to fouling (Rt), recoverable flux (Rr), and non-recoverable flux (Rir). The pure PES membrane has denser pore characteristics as confirmed by the surface SEM images and high HA rejection results. The dense surface of the membrane causes the HA particle to easily accumulate on the membrane surface, resulting in a total flux loss (Rt) of 50.3%. Membrane cleaning could only recover 5.3% of those flux-loss due to the strong interactions between the hydrophobic HA particles and the surface of the PES membrane [28,32]. The modified membranes, especially those polymerized at shorter duration, suffer higher total flux loss than those with higher dopamine concentration and more extended dipping periods. This is because the presence of dopamine and PDA layer increases the membrane hydrophilicity (Figure 3), which improves the membrane cleaning effectivity as observed from the Rr and Rir data in Figure 6.

### 3.5. UV Resistance Performance

The SEM images in Figure 7 indicate a degree of the protective effect of polymerization on the resistivity of the membrane upon exposure to UV-irradiation. The dopamine concentration and polymerization time have a positive impact on protecting the membrane from UV radiation. This can be observed through changes in membrane surface morphology obtained from analysis using SEM. To investigate the influences of modifications and their variables on the membrane resistance to UV irradiation, the UV irradiation test was carried out following the previously reported method [28]. SEM analysis was carried out on membranes P, P-D5-a, P-D5-c, and P-D36-c without UV exposure and after seven days of radiation to UV.

Figure 7 shows that, after seven days of UV exposure, the pure PES membrane was severely damaged due to radical-induced photodegradation. Almost no part of the surface retained the dense top layer of the membrane. After modification using dopamine at low concentrations and short polymerization times, the damage is still visibly bad but slightly better if compared to that of the radiated pristine PES membrane. By increasing dopamine and polymerization time, damage to the membrane surface is seen to be reduced. This is because, at high dopamine concentrations and long polymerization times, the PDA layer formed on the membrane due to the polymerization triggered by the Tris-HCl solution is also getting thicker. The PDA layer formed on the membrane surface reduces the morphological damage caused by UV radiation, as reported in our previous research [28], even though the results are not as excellent as the membrane modified with a surface coating. It is worth noting the test was done under extreme exposure to accelerate the degradation process. In the real application, the UV intensity and duration may be less and the damage would not be as severe as demonstrated in Figure 7. It is worth noting that SEM images pose inherent limitations because of the small area of the imaging. However, the findings in Figure 7 are representative of images from homogeneous surface properties over a few samples. Nontheless, we include the mechanical properties results which confirm the SEM findings regarding the membrane stability after treatment (Figure 8 and Figure 9). We believe that the prolonged duration or too-high intensity of the UV exposure led to less demonstrative findings from the SEM image.

### 3.6. Effects on Mechanical Strength

The mechanical properties of the membrane were evaluated in terms of tensile strength as shown in Figure 8. In general, there is a decrease in tensile strength in all dopamine-modified membranes, most likely due to a more porous structure rich with macro-void as detailed elsewhere [38,39]. However, immersing the blended membrane in the alkaline solution brings back the tensile strength of the membrane. This is attributed to the adhesive property of the PDA film, which strengthens the membrane structure altogether [32,40].

The tensile strength measurements were taken before and after UV radiation to investigate the effect of UV irradiation on the mechanical strength of all membranes. UV radiation was carried out for a week where the data of tensile strength were taken on the first day (24 h of radiation), the third day (72 h of radiation), and the seventh day (120 h of radiation). The measurement results are presented in Figure 9.

The tensile strength is also affected by the membrane thickness and porosity. Increasing dopamine concentration in the dope solution resulted in an increase in membrane porosity and a decrease in thickness, in which P, P-D0-a, P-D0-b, and P-D0-c had porosity of 2.42 ± 0.24, 8.87 ± 0.33, 10.02 ± 0.17, and 18.63 ± 0.29%, while the thicknesses of 1.2, 0.7, 0.6, and 0.5 mm.

After exposure to UV radiation from the pristine membrane and after modification, the mechanical strength was compared with the nonradiated membranes in the form of retention. In Figure 10, the vertical axis shows the retention of the tensile strength (%), which is the tensile strength after UV exposure divided by tensile strength before UV exposure. Data in Figure 10 demonstrate that pristine PES membranes experienced a significant decrease after UV radiation, losing up to 65% of the initial tensile strength after seven days of exposure. A similar occurrence is also seen in dopamine-blended membranes without polymerization (P-D0-a; P-D0-b; P-D0-c). On these membranes, the decrease in tensile strength is even more significant than that of the original PES membrane, especially for the ones with the addition of high concentrations of dopamine. This significant decrease is due to the porous morphological influence of the membrane, which makes the membrane matrix weaker and more easily degraded due to radiation [41]. The mechanical degradation of the membrane is caused by the morphological damage of the membrane by free radicals from UV radiation. Through several stages, it degrades the polymer structure of the membrane through abstraction of the H atom to the cutting of the main polymer chain [41,42]. Significant mechanical strength is maintained by membranes after polymerization, even for the most prolonged UV exposure (90%), demonstrating the effectiveness of the polydopamine in providing UV resistant property on the membrane.

### 3.7. Free Radical Scavenging Mechanism of Polydopamine

By adapting the schematic diagram of the stabilization mechanism using a phosphite antioxidant reported by Voigt and Todesco [43], Figure 11 presents the proposed mechanism of UV protection by PDA. In more detail, the mechanism of free radical scavenging by PDA is as follows: a polymer (RH), when exposed to radiation (UV light), will form radicals (R•). These radicals then react with O_2_ (existing in the environment) to generate peroxy radicals (ROO•). This peroxy radical is unstable and subtracts hydrogen atoms from the polymer chains (RH) to form hydroperoxides (ROOH) and more polymer radicals (R•). This is the stage where the photodegradation starts occurring and causes the scission of polymer chains, which yield to the change in chemical membrane components and morphological structure [44].

Polydopamine consists of a long chain of catecholamine rich in hydroxyl (-OH) and amine (-NH) groups. Both functional groups are electron-donating. Immersing the dopamine-blended membrane into Tris-solution triggered a thin polydopamine layer on the membrane surface. The layer then protects the polymer matrix from being photo-degraded by quenching/deactivating the UV-induced polymer free radicals through the electron donor process. During the propagation stage of photo-degradation (abstraction of a hydrogen atom from polymer chains by radicals to form ROOH), the radicals obtain hydrogen by reacting with the phenolic rings from the polydopamine instead of snatching the hydrogen from the polymer chain (RH). The formed ROOH reacts further with the scavenger groups from polydopamine and breaks down as inactive compounds such as ROH and H_2_O. These scavenging activities stop the radicals from abstracting the hydrogen from the polymer chain leading to stabilization, which eventually restricts photodegradation of the polymer [45].

## 4. Conclusions

The two-step dopamine-to-polydopamine modification on the PES-based ultrafiltration membranes has proven to increase the fouling and UV-irradiation resistance of the resulting membranes. The treatment step, concentration of dopamine, and duration of dipping in Tris solution affect the properties of the resulting membranes. Blending dopamine at a higher concentration improves PWF by more than threefold that of the original PES membrane (15 vs. 50 L/m^2^.h). However, extended dipping in Tris alkaline buffer solution triggers an overgrowth of the polydopamine layer on the membrane surface that covers surface pores, leading to the decline of CWF. The presence of polydopamine enhances the hydrophilicity due to the enrichment of hydrophilic catechol moieties, which leads to better anti-fouling performance, as seen from the resulting FRR of 96%. Moreover, the polydopamine film simultaneously improves the membrane resistance to UV by minimizing the occurrence of photodegradation, as proven by the retention of tensile strength.

## Figures and Tables

**Figure 1 polymers-12-02051-f001:**
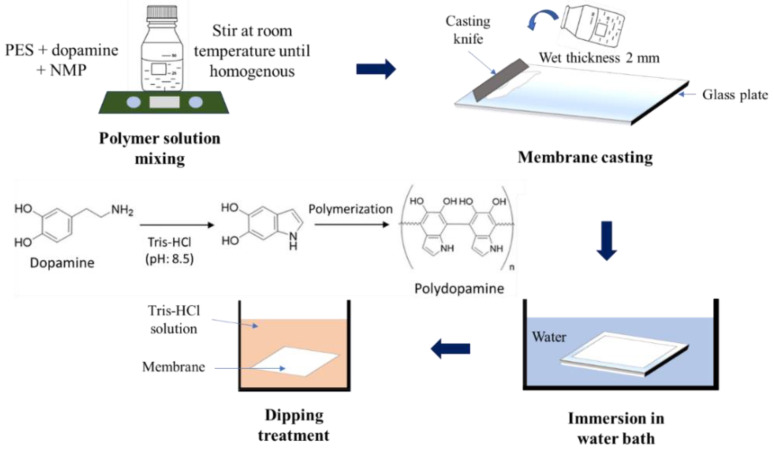
Illustration of membrane fabrication method showing the step of membrane fabrication including dopamine polymerization.

**Figure 2 polymers-12-02051-f002:**
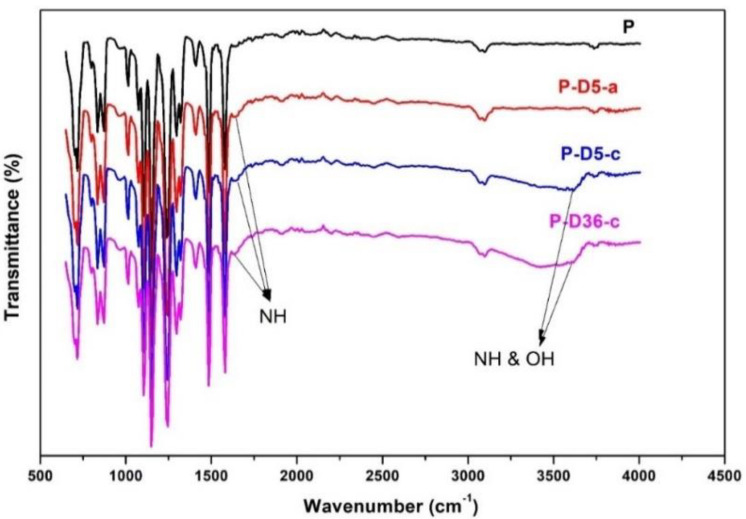
The changes in chemical functional groups of PES membranes before and after modification.

**Figure 3 polymers-12-02051-f003:**
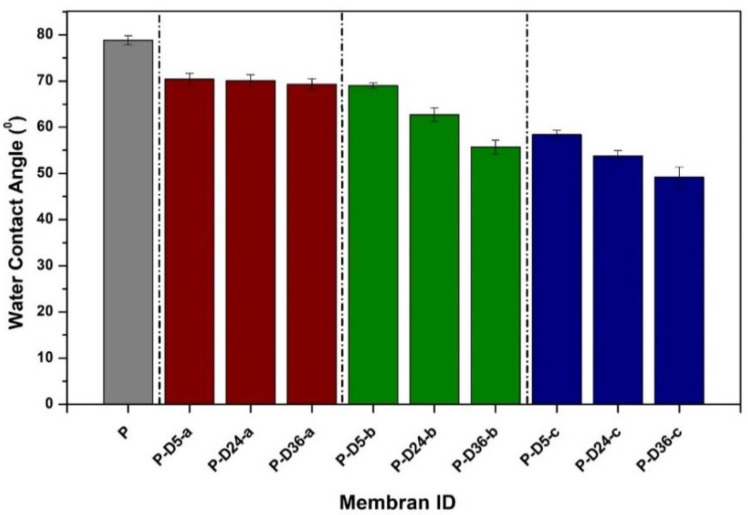
Water contact angle of the prepared membranes. The data are presented as average ± standard deviation from *n* = 10.

**Figure 4 polymers-12-02051-f004:**
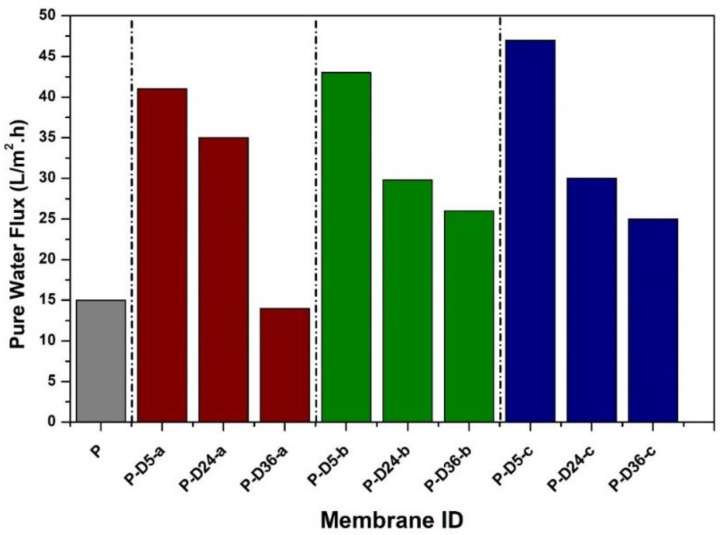
PES membrane pure water flux performance before and after modification with dopamine at various concentrations and dipping times.

**Figure 5 polymers-12-02051-f005:**
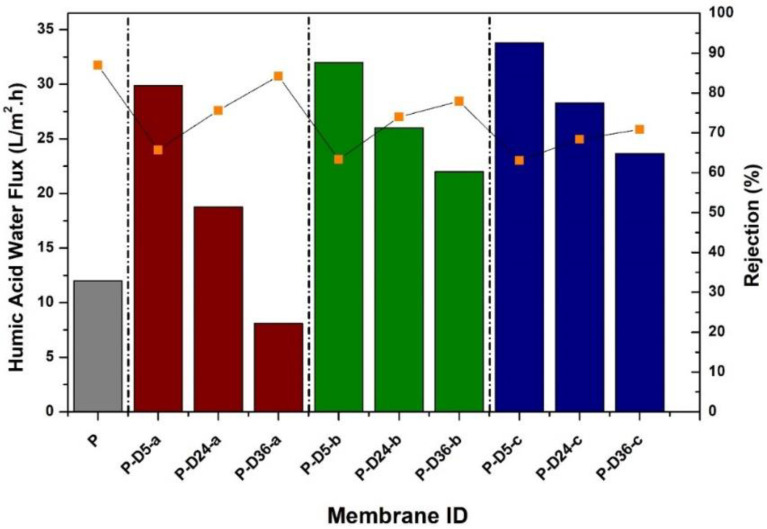
Humic acid solution water flux and rejection performances of the PES membrane without and with dopamine modification at various concentrations and dipping times.

**Figure 6 polymers-12-02051-f006:**
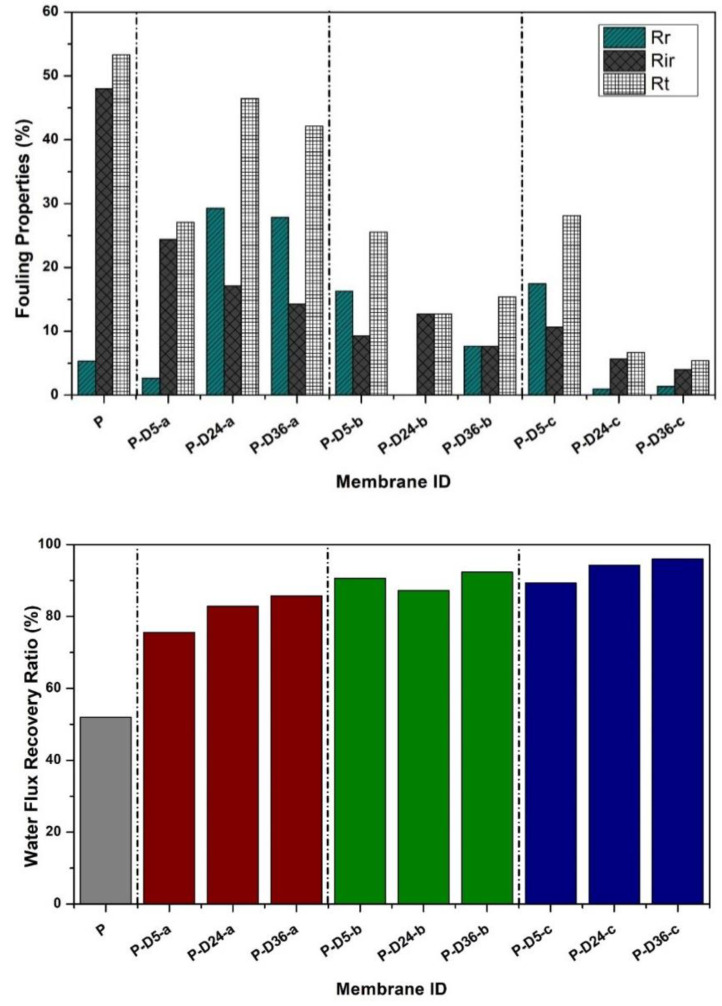
Effect of modification on membrane antifouling performance in terms of (top) Rr, Rt, Rir, and (bottom) the FRR.

**Figure 7 polymers-12-02051-f007:**
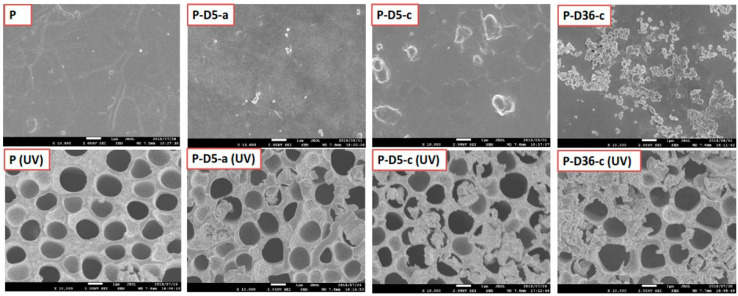
Morphological change of original and modified PES membranes before (**top row**) and after (**bottom row**) seven days of UV exposure.

**Figure 8 polymers-12-02051-f008:**
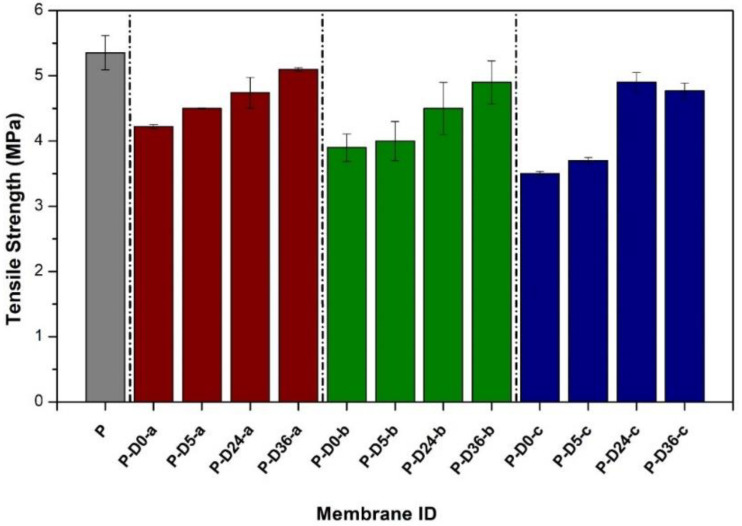
Tensile strength of original PES membranes and dopamine modified PES membranes at various concentrations and polymerization times. The data are presented as average ± standard deviation from *n* = 5.

**Figure 9 polymers-12-02051-f009:**
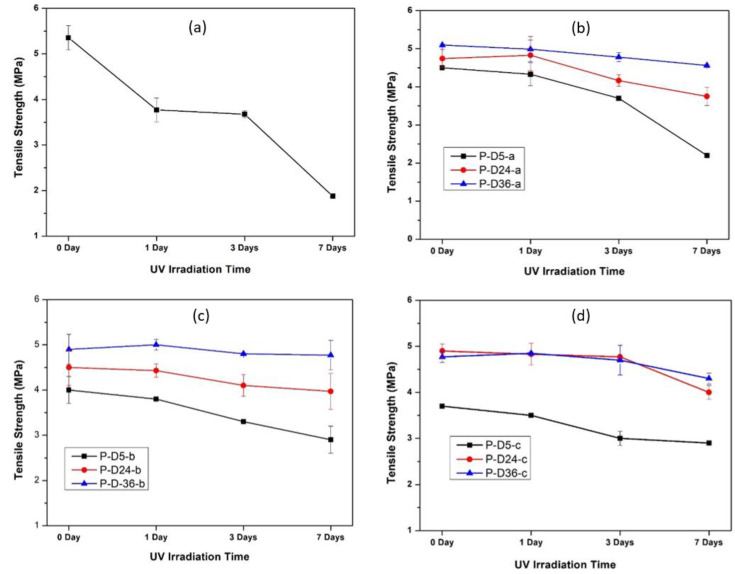
Tensile strength of original and modified membranes before and after UV exposure: (**a**) pristine PSF, (**b**) 0.5% dopamine loading at different polymerization times, (**c**) 2.0% dopamine loading at different polymerization times, and (**d**) 4.0% dopamine loading at different polymerization times. The data are presented as average±standard deviation from *n* = 5.

**Figure 10 polymers-12-02051-f010:**
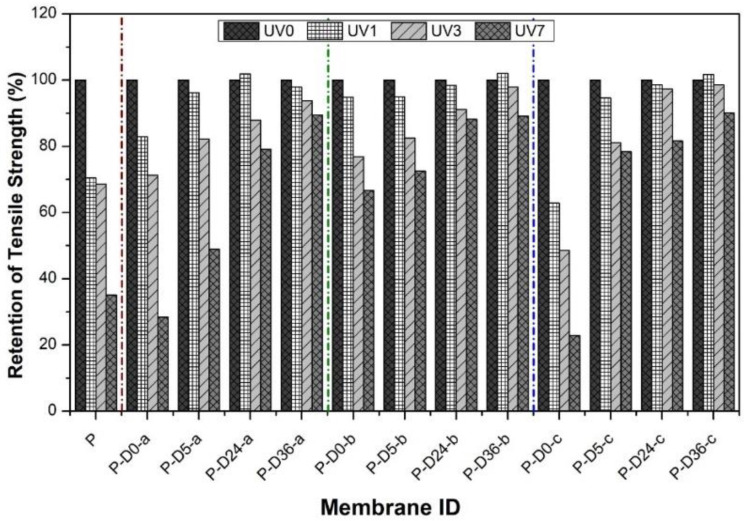
Retention of tensile strength of membranes before and after UV exposure.

**Figure 11 polymers-12-02051-f011:**
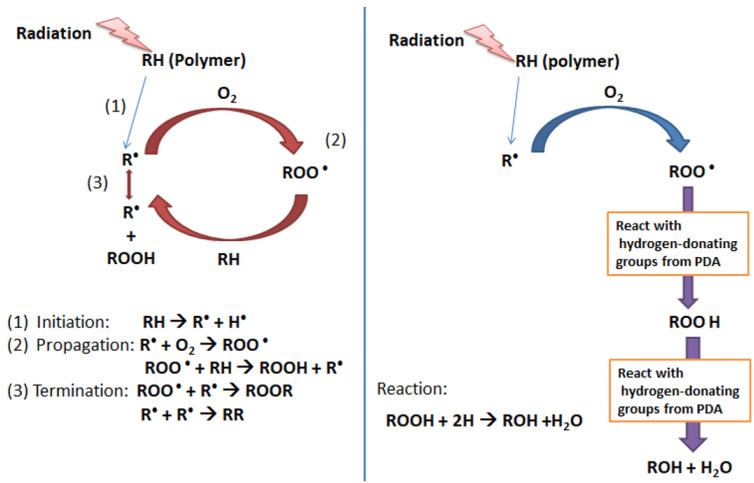
Proposed scavenging mechanism of a UV-induced radical by PDA: (**left**) photodegradation mechanism of the polymer and (**right**) proposed mechanism of free-radical scavenging by polydopamine.

**Table 1 polymers-12-02051-t001:** Composition of polymer solution for membrane fabrication.

No.	Membrane ID	Concentration (%)			Polymerization Time (hour)
		PES	NMP	Dopamine	
1	P	17.5	82.5	0	0
2	P-D0-a	17.5	82	0.5	0
3	P-D5-a	17.5	82	0.5	5
4	P-D24-a	17.5	82	0.5	24
5	P-D36-a	17.5	82	0.5	36
6	P-D0-b	17.5	80.5	2	0
7	P-D5-b	17.5	80.5	2	5
8	P-D24-b	17.5	80.5	2	24
9	P-D36-b	17.5	80.5	2	36
10	P-D0-c	17.5	78.5	4	0
11	P-D5-c	17.5	78.5	4	5
12	P-D24-c	17.5	78.5	4	24
13	P-D36-c	17.5	78.5	4	36

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
