# Peer review of "Two-Step Dopamine-to-Polydopamine Modification of Polyethersulfone Ultrafiltration Membrane for Enhancing Anti-Fouling and Ultraviolet Resistant Properties"

_polymers, 2020, doi:10.3390/polym12092051_

Round 1

Reviewer 1 Report

This manuscript submitted by Mulyati reports the modification of polyethersulfone ultrafiltration membrane for improved physicochemical properties, such as anti-fouling and ultraviolet resistance. The paper is organized well and it is easy for reading and understanding. The results were analyzed clearly and support their conclusion. This manuscript can be recommended for publication after addressing the following minor issues.

  • In literature, most papers about PDA coating were using a one-step method by incubating the substrates in dopamine solution. However, this work adopted a two-step method. I guess the authors would like to avoid the overcoating problem? If so, is it possible to use a low concentration of dopamine in a one-step strategy? Please clarify this.
  • In 3.5. UV Resistance Performance section, the authors utilize SEM to evaluate the stability of membrane under UV irradiation. Any quantitative characterization to confirm this, such as absorbance? It is because SEM images in such as a confined area cannot tell the overall structure of the membrane after treatment.
  • The authors should clarify what is the N number for deviations in Figure 3, 8, and 9.
  • I am surprised that the seminal work (10.1126/science.1147241) and some recent reviews (10.1021/acs.accounts.0c00150; 10.1039/C9CS00849G) are not cited in this manuscript.

Reviewer 2 Report

This work by Mulyati et al, reported the modification of PES ultrafiltration membrane by two-step dopamine-to-polydopamine. The resultant membrane showed enhanced anti-fouling and ultraviolet resistant properties. However, I do not think it can be accepted in Polymers. 1. Lack of novelty is a major problem. Very similar research has been reported ( Journal of Water Process Engineering 2019, 28, 293-299). Also, there are lots and lots of PDA-modified UF membranes for anti-fouling and anti-UV. Therefore, the authors need to present the difference/novelty of the research, for example, in the Introduction section. 2. This manuscript lacks many key experiemtns for the evaluation of UF membranes. For example, the thickness, pore size, porosity. Also, From Fig. 7, why the membranes look no pores on the surface before UV exposure? what is the morphology of the cross-section of the membrane? 3. The authors immersed the casting membrane to water, then to Tris buffer. Why did not directly immerse the casting membrane to Tris buffer for hours?

Round 2

Reviewer 1 Report

N.A.

Reviewer 2 Report

After evaluating the comments from the authors and their revised manuscript, I think it can be published now.